# Ordering a Normal Diet at the End of Surgery—Justified or Overhasty?

**DOI:** 10.3390/nu10111758

**Published:** 2018-11-14

**Authors:** Fabian Grass, Martin Hübner, Jenna K. Lovely, Jacopo Crippa, Kellie L. Mathis, David W. Larson

**Affiliations:** 1Division of Colon and Rectal Surgery, Mayo Clinic, 200 First Street SW, Rochester, MN 55905, USA; grass.fabian@mayo.edu (F.G.); crippa.jacopo@mayo.edu (J.C.); mathis.kellie@mayo.edu (K.L.M.); larson.david2@mayo.edu (D.W.L.); 2Department of Visceral Surgery, Lausanne University Hospital CHUV, 1011 Lausanne, Switzerland; 3Hospital Pharmacy Services, Mayo Clinic, 200 First Street SW, Rochester, MN 55905, USA; lovely.jenna@mayo.edu

**Keywords:** nutrition, enhanced recovery, colorectal

## Abstract

Early re-alimentation is advocated by enhanced recovery pathways (ERP). This study aimed to assess compliance to ERP-set early re-alimentation policy and to compare outcomes of early fed patients and patients in whom early feeding was withhold due to the independent decision making of the surgeon. For this purpose, demographic, surgical and outcome data of all consecutive elective colorectal surgical procedures (2011–2016) were retrieved from a prospectively maintained institutional ERP database. The primary endpoint was postoperative ileus (POI). Surgical 30-day outcome and length of stay were compared between patients undergoing the pathway-intended early re-alimentation pattern and patients in whom early re-alimentation was not compliant. Out of the 7103 patients included, 1241 (17.4%) were not compliant with ERP re-alimentation. Patients with delayed re-alimentation presented with more postoperative complications (37 vs. 21%, *p* < 0.001) and a prolonged length of hospital stay (8 ± 7 vs. 5 ± 4 days, *p* < 0.001). While male gender (odds ratio (OR) 1.24; 95% confidence interval (CI) 1.04–1.32), fluid overload (OR 1.38; 95% CI 1.16–1.65) and high American Society of Anaesthesiologists (ASA) score (OR 1.51; 95% CI 1.27–1.8) were independent risk factors for POI, laparoscopy (OR 0.51; 95% CI 0.38–0.68) and ERP compliant diet (OR 0.46; 95% CI 0.36–0.6) were both protective. Hence, this study provides further evidence of the beneficial effect of early oral feeding after colorectal surgery.

## 1. Introduction

American and European guidelines advocate early oral fluid and solid intake after surgery to prevent a perioperative decline in nutritional status [1,2,3]. Early re-alimentation is embedded in a multimodal concept in enhanced recovery pathways (ERP), aiming to decrease surgery-associated physiologic stress response [4,5]. ERPs have been widely adopted as standard of care due to their potential to decrease postoperative complications and, as a consequence, length of stay and costs [6,7,8]. While prevailing debates focus on ideal intraoperative fluid and postoperative pain management, the nutritional aspect may be undervalued and considered as “given” [9]. However, recent publications demonstrated early resumption of normal diet to be a particular challenge [10,11], and surgeons at time make intuitive decisions based on experience to bypass ERP recommendations for early feeding by prescribing postoperative orders to deviate from this standard.

The aim of this present study was to address the question how many patients were assigned to pathway-compliant re-alimentation and whether surgeons’ appraisal to withhold a normal diet was justified.

## 2. Materials and Methods

### 2.1. Patients

This study was conducted as part of a global enhanced recovery quality improvement project and approved by the institutional review board. Demographic and surgical data were retrieved from the prospectively-maintained ERP database over the 6-year study period (2011–2016). ERP was started in November 2009, fully implemented in the Department of Colon and Rectal Surgery, Mayo Clinic, Rochester, MN, USA, a tertiary academic facility, in 2010, and became a Division standard as of January 2011. The contents of this database have been described previously in several institutional series with focus on compliance and outcome after minimally invasive colon and rectal surgery [12,13,14,15,16].

All consecutive colorectal procedures, including colectomies, rectal procedures and ostomy procedures (including Hartmann reversals) were included. Other procedures included small bowel resections (including ileal pouch anal anastomosis (IPAA) surgery for inflammatory bowel disease), as well as rectal prolapse repairs. Malnutrition was assessed through measurements of preoperative serum albumin within 90 days of surgery and stratified according to institutional thresholds (hypoalbuminemia defined as <3.5 mg/dL). Total amount of intravenous (IV) fluids (crystalloids, blood products, albumin and volume expander) at postoperative day (POD) 0 (intraoperative and until midnight) were retrieved from electronic medical records. A threshold of 3 L was defined (Hübner, BJS Open in press) for the purpose of further comparisons. Further, weight at POD 1, 2 and at discharge and ingested oral liquids through POD 0–2 were quantified by ward nurses.

### 2.2. Enhanced Recovery Pathway

Details of the pathway have been described previously [12]. Briefly, the institutional ERP focuses on systemic postoperative nausea and vomiting (PONV) prophylaxis, a single injection intrathecal for analgesia, pre- and postoperative opioid-sparing multimodal analgesia including non-steroidal anti-inflammatory drugs (NSAID) for patients meeting criteria, stringent intraoperative fluid administration, and early postoperative mobilization and re-alimentation patterns at 4 h post-surgery. As a general rule, nasogastric tubes were removed by the end of the surgical procedure.

### 2.3. Outcomes/Study Endpoints

The primary endpoint was postoperative ileus (POI), defined as re-insertion of a nasogastric tube [17]. The operating surgeon decided on the re-alimentation pattern at the end of the surgical procedure by signing the computed ERP order. Patients assigned to ERP-compliant re-alimentation resumed a normal low-fiber or diabetic diet 4 h after surgery and benefit of a flexible, dedicated and patient-driven “food service” without dietary or caloric restrictions (except for uncooked fruits and vegetables). Through this approach, patients get to call, choose and eat whatever they desire, without caloric restrictions. The diet is complemented by nutritional supplements (protein drinks) if indicated based on nutritional assessment.

If, according to surgeons’ appraisal, standard ERP re-alimentation was not desired, orders were written, which resulted in the patient dropping out of the intended ERP re-feeding policy to follow an individual, delayed re-alimentation pattern during the subsequent postoperative days.

Clinical outcome was evaluated in-hospital and until 30 days postoperatively, and compared between the two groups (ERP diet vs. individual diet). Overall complications were classified according to Clavien Dindo grade I-V [18]. Specific complications were further assessed: bleeding complications (need for transfusion at POD 0–2), surgical site infections (SSI) needing either surgical, percutaneous or negative pressure wound therapy, clinically or radiologically confirmed anastomotic leaks and POI.

### 2.4. Statistical Analysis

Descriptive statistics for categorical variables were reported as frequency (%), while continuous variables were reported as mean (standard deviation) or median (interquartile range, IQR) as appropriate. Chi-square test was used for comparison of categorical variables. All statistical tests were two-sided, *p* value < 0.05 was used to indicate statistical significance. All significant demographic and surgical risk factors for POI (including the confounder ERP diet order) upon univariate analysis were entered into a multivariate logistic regression (based on a probit regression model) to provide adjusted estimations of the odds ratio (OR). Data analysis was performed with the Statistical Software for the Social Sciences SPSS Advanced Statistics 22 (IBM Software Group, 200 W. Madison St., Chicago, IL, USA).

## 3. Results

### 3.1. Patients

A total of 7103 patients were enrolled. Of these, 1241 patients (17.4%) were deprived of the standardized re-alimentation pattern as set by the institutional ERP and dropped out to undergo individualized re-alimentation, upon the surgeons’ discretion. The demographic and surgical characteristics of both groups are displayed in Table 1.

Weight gain in patients assigned to ERP diet vs. patients assigned to individual diet was as follows: 2.3 ± 1.6 vs. 3.2 ± 6.7 kg at POD 1 (*p* = 0.087), 2.5 ± 1.7 vs. 3.6 ± 6.2 kg at POD 2 (*p* = 0.022) and 1.1 ± 1.7 vs. 1.9 ± 6.4 kg at discharge (*p* = 0.125). Ingested oral liquids were quantified as follows: 510 ± 390 vs. 330 ± 320 mL at POD 0, 1400 ± 660 vs. 1090 ± 690 mL at POD 1 and 1120 ± 640 vs. 1020 ± 640 mL at POD 2 (all *p* < 0.001).

### 3.2. Outcome

All outcomes differed significantly among the two groups, as illustrated in Figure 1 (any complication, bleeding complication, POI, anastomotic leak, SSI and readmission all *p* < 0.001, reoperation rate 6.8% in patients assigned to ERP re-alimentation pattern vs. 9.2% in patients assigned to individual re-alimentation pattern, *p* = 0.003). Mean length of stay was shorter in patients assigned to the ERP diet (4.9 ± 5.3 vs. 7.5 ± 8 days, *p* = < 0.001).

### 3.3. Postoperative Ileus

Postoperative ileus occurred in 714 patients (10%) at the 4th (IQR 3–6) postoperative day.

Male gender (OR 1.24; 95% CI 1.04–1.32, *p* = 0.01), fluid overload (OR 1.38; 95% CI 1.16–1.65, *p* < 0.001) and high American Society of Anaesthesiologists (ASA) score (OR 1.51; 95% CI 1.27–1.8, *p* < 0.001) were independent risk factors for POI, while laparoscopy (OR 0.51; 95% CI 0.38–0.68, *p* < 0.001) was a protective factor. Further, ordering of the ERP-compliant normal diet was associated with decreased POI (OR 0.46; 95% CI 0.36–0.6, *p* < 0.001, Figure 2).

## 4. Discussion

In 83% of patients, a normal postoperative diet according to the institutional ERP was prescribed by the operating surgeon at the end of the procedure, and this intuitive choice seemed to be motivated by patient constitution and surgical difficulty. Patients who followed the ERP-compliant normal re-alimentation pattern experienced fewer overall and specific postoperative complications and a shorter length of stay. Thus, the present study provides further evidence of the beneficial impact of early postoperative re-alimentation after colorectal surgery and does not support the surgeons’ intuitive decision to bypass early feeding.

The beneficial impact of early postoperative nutrition has been demonstrated by several meta-analyses of randomized controlled studies and was endorsed by nutritional societies and enhanced recovery guidelines [19,20,21]. A landmark study by Hiesmayr et al. revealed a significant association between decreased food intake and in-hospital mortality through a multi-national cross-sectional survey, demonstrating further that more than half of the patients did not eat the full meal provided by the hospital [22]. These results of increased morbidity and decreased food intake in over half of patients were confirmed by a recent observational study of patients treated within an ERP [10].

The present study emphasizes the importance of an early normal diet through the association with decreased complications and length of stay. Hence, the decision to withhold or postpone a normal diet is not supported by our results and may even be a contributing factor for postoperative adverse events. Different patient- and disease related factors such as age, ASA score, malnutrition and malignancy, but also factors indicating surgical difficulty (through surrogates surgical duration, open approach and perioperative fluid administration) arguably impacted the surgeons’ decision to deviate from the standardized pathway (Table 1). However, this series seems to suggest that surgeon decision-making based on patient and operative factors is not a reasonable decision aid to make appropriate patient-centered decisions about early feeding. This surgeon “gut feeling” consciously, or also unconsciously, taking into account all these different potential risk factors related to the patient and the procedure appears to lead to poorer patient outcome and may represent conformational bias which is a none contributor to poor decision making [23,24,25]. Even though the observational design of this study does not allow drawing causative conclusions, the present results suggest good surgical judgement includes following ERP based re-alimentation pattern despite patient and operative factors.

As it is of utmost importance, the concept of early enteral nutrition needs to be embedded in a multimodal nutritional approach. ERP may have a positive impact on the adoption of standardized perioperative nutrition care practices, in particular by increasing detection and timely treatment of malnutrition [26]. Exercise, nutrition and anxiety reduction further complement ERP and facilitate return to baseline activities of daily living including normal nutrition [27]. Thus, prehabilitation strategies including nutritional support need to be considered as complementary to the concept of early postoperative re-alimentation [28,29]. Besides preoperative conditioning and the aforementioned perioperative nutritional concept, stringent fluid management and minimally invasive surgery, which were both preferentially used in the group receiving the ERP-compliant normal diet, arguably further contributed to improved postoperative outcomes in the present cohort.

The present study has several limitations due to its observational design. Surgical interventions and diseases were heterogeneous in this unselected cohort. No cause-effect patterns are evident, and different patient- and disease- related factors, as well as early postoperative complications within 24 h of surgery may have an impact on re-alimentation patterns. Actual oral intake was not quantified in the present study, it was assumed that patients assigned to the ERP-compliant re-alimentation pattern ate most of their meal. Recent studies found compliance rates to hospital diet of up to 60% [10,22]. The present institution offers a more flexible approach with dedicated, patient-driven food service; compliance rates and caloric intake in this setting need to be further evaluated. Nevertheless, this situation analysis provides useful information on the feasibility of the current institutional nutritional standard ERP orders and the specific role of the individual surgeon.

## 5. Conclusions

In conclusion, compliance with early ERP re-alimentation ordering standards likely contributed to improved postoperative outcomes after elective colorectal procedures in this large cohort. Surgeons intuitively decided to deviate from the intended pathway, considering demographic and surgical factors. However, the independent protective effect of the ERP-compliant realimentation order does not support the surgeons’ decision to delay early feeding based on these criteria.

## Figures and Tables

**Figure 1 nutrients-10-01758-f001:**
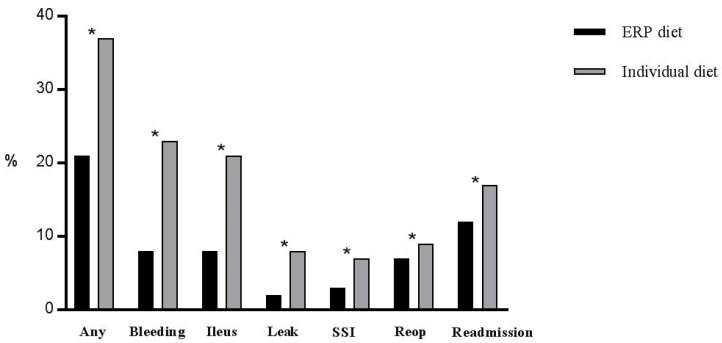
Outcome. Postoperative complications in patients assigned to the ERP re-alimentation pattern (*n* = 5862) and patients assigned to an individualized re-alimentation pattern (*n* = 1241). Any complication: Clavien grade I-V, ERP—enhanced recovery pathway, SSI—surgical site infection, reop—reoperation. ***** indicates statistical significance (*p* < 0.05).

**Figure 2 nutrients-10-01758-f002:**
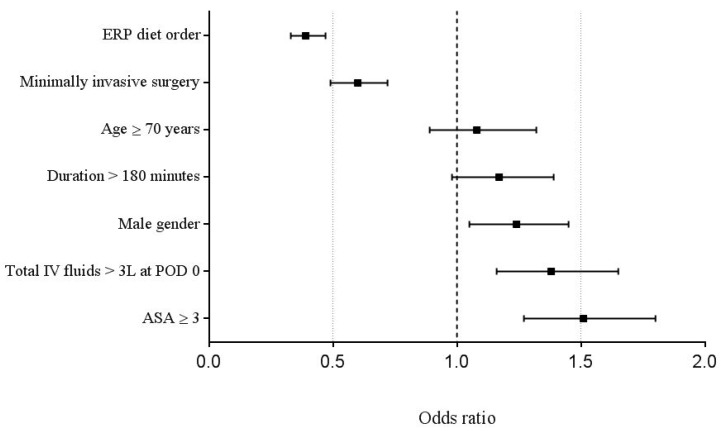
Multivariable analysis. Multivariable analysis of univariate demographic and surgical items (*p* < 0.05) associated with POI. An Odds ratio of >1 indicates increased risk of POI. ASA—American Society of Anaesthesiologists, ERP—enhanced recovery pathway, IV—intravenous, POI—postoperative ileus. Odds ratio, 95% Confidence Interval.

**Table 1 nutrients-10-01758-t001:** Demographic and surgical parameter.

	All (*n* = 7103)	ERP Diet (*n* = 5862)	Individual Diet (*n* = 1241)	*p*
Age (years) (mean ± SD)	53 ± 18	53 ± 18	56 ± 18	<0.001
Age > 70 years (%)	1471 (21)	1177 (20)	294 (24)	0.005
Male gender (%)	3614 (51)	2995 (51)	619 (50)	0.453
BMI (kg/m^2^) (mean ± SD)	27 ± 6.9	27 ± 6.9	27.2 ± 6.7	0.349
ASA Group ≥ 3 (%)	1883 (29)	1382 (26)	501 (42)	<0.001
Diabetes Mellitus (%)	616 (9)	487 (8)	129 (10)	0.02
Preoperative albumin (g/dL) (%)	4 ± 0.6	4 ± 0.6	3.8 ± 0.7	<0.001
<3.5 g/dL	563/2706 (20)	393/2142 (18)	170/564 (30)	<0.001
Malignancy (%)	3863 (54)	3131 (53)	732 (59)	<0.001
Perioperative fluid management				
Total intraoperative fluids	2380 ± 1790	2250 ± 1510	3020 ± 2670	<0.001
Total fluids POD 0	3050 ± 1910	2900 ± 1650	3750 ± 2750	<0.001
Fluids POD 0 > 3 L	2965 (42)	2318 (40)	647 (52)	<0.001
Minimally invasive approach (%)	2613 (37)	2311 (39)	302 (24)	<0.001
Procedure				
Colon resection (%)	3836 (54)	3151 (54)	685 (55)	0.354
Rectal resection (%)	913 (13)	762 (13)	151 (12)	0.427
Ostomy procedure (%)	994 (14)	839 (14)	155 (13)	0.093
Other (%)	1360 (19)	1110 (19)	250 (20)	0.325
Operation duration (min) (mean ± SD)	170 ± 100	160 ± 90	200 ± 150	<0.001
>180 min (%)	2709 (38)	2161 (37)	548 (44)	<0.001

Baseline demographic and surgical parameters of patients assigned to the ERP re-alimentation pattern (*n* = 5862) and patients assigned to an individualized re-alimentation pattern (*n* = 1241). ASA—American Society of Anaesthesiologists, BMI—body mass index, ERP—enhanced recovery pathway, POD—postoperative day, SD—standard deviation. Age and BMI are presented as mean ± standard deviation. All others are frequency with percentage. Bold characters indicate significant values (*p* < 0.05).

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
