# Peer review of "Ordering a Normal Diet at the End of Surgery—Justified or Overhasty?"

_nutrients, 2018, doi:10.3390/nu10111758_

Round 1

Reviewer 1 Report

Grass and colleagues report on their study of 7103 patients where they studied the benefit of an ERP early oral feeding diet versus traditional delayed feeding diet.  The main conclusion is that compliance with early ERP re-alimentation ordering standards was associated with improved postoperative outcomes after elective colorectal procedures.  This is an interesting and well written paper, however it would be strengthened if the authors could address the following issues:

A problem with the study is that they did not determine to what extent the patients actually received the early diet (ie just because the surgeon ordered the early diet did not mean the patient received it) - please discuss in the context of your results.

There are other reasons to explain your improvements in outcome such as higher rates of MIS in your ERP group.  There is also lower percentages of ASA Group ≥ 3 and diabetics.  Please elaborate more on this within the discussion.

Author Response

Grass and colleagues report on their study of 7103 patients where they studied the benefit of an ERP early oral feeding diet versus traditional delayed feeding diet.  The main conclusion is that compliance with early ERP re-alimentation ordering standards was associated with improved postoperative outcomes after elective colorectal procedures.  This is an interesting and well written paper; however it would be strengthened if the authors could address the following issues:

A problem with the study is that they did not determine to what extent the patients actually received the early diet (ie just because the surgeon ordered the early diet did not mean the patient received it) - please discuss in the context of your results.

This is an important comment and indeed represents a limitation of our study. We tried to emphasize this in the limitation section. We further expanded on this important issue at the end of the discussion.

There are other reasons to explain your improvements in outcome such as higher rates of MIS in your ERP group. There is also lower percentages of ASA Group ≥ 3 and diabetics.  Please elaborate more on this within the discussion.

Indeed, surgeons preferably prescribed a normal diet after MIS and in seemingly less sick patients. We agree that ERP diet is one of several factors that contributed to favorable postoperative outcomes and acknowledged this further at the end of the discussion. However, we believe that considering the independent protective effect of early realimentation, surgeons should not decide based on demographic and surgical factors and intuition to withhold a normal ERP diet from the patient.

Reviewer 2 Report

Dear editor,

This is a well written paper from well known “fast track” specialists.

The strength of this study is the large sample size and an overall well-defined cohort. The weakness is what always is the problem with retrospective cohorts from large databases, missing data and the problem with causality.

In summary, I think this paper should be considered for publication.

Major concerns

A major drawback is that the authors cannot show the mean number of calories that were received in both groups. Thus, compliance to both the ERP nutritional item and individual diet is missing. Thus, in theory, we do not know for sure that the ERP patients had more calories than the individual diet group.

However, retrospective data on nutritional intake are often uncertain and difficult to analyze. Therefore, since we know that compliance to nutritional intake is approximately 60%-70% in dedicated ERAS/ERP centers, it would have been a great help with a definition of the current ERP protocol. What was the standard re-alimentation protocol other than resumption of low-fiber or diabetic nutrition as mentioned in the article? Could the authors add the amount of calories, nutritiondrinks etc POD 0-3 (ie what patients should receive according to the protocol)?

Causuality: We know that the surgeon decided in the end of operation who goes for ERP nutrition and who goes for individual nutrition (that is a strength with this study). However, individually treated patients had a way higher morbidity than the ERP patients and would probably have suffered from more complications regardless of nutritional protocol (confirmed by the multivariate analysis). No matter how you try in multivariate analysis, it is difficult to adjust for this, and find true independent predictors. Therefore, I think it is important to show exactly how the analysis was performed.

Which variables were used? The seven that are shown in Figure 2? Does probit regression mean that these variables were kept in the model all the time? That the current OR for each predictor holds for adjustments with the other 6 in the model?

Minor concerns

Male gender seems to be an independent predictor for POI (at least when you look at the Figure) but 1.24 (0.89-1.32) is not significant as stated in abstract and text in the article.

I think the conclusion is to firm (considering the above mentioned in major concerns). Maybe the authors can make it softer??

Author Response

This is a well written paper from well known “fast track” specialists.

The strength of this study is the large sample size and an overall well-defined cohort. The weakness is what always is the problem with retrospective cohorts from large databases, missing data and the problem with causality.

In summary, I think this paper should be considered for publication.

Major concerns

A major drawback is that the authors cannot show the mean number of calories that were received in both groups. Thus, compliance to both the ERP nutritional item and individual diet is missing. Thus, in theory, we do not know for sure that the ERP patients had more calories than the individual diet group.

We agree that the lack of compliance data represents a limitation of the study. Even if we assume that compliance was not 100% but rather 60% according to recent evidence, we believe that patients within the ERP-compliant realimentation pathway likely received more calories compared to patients in whom realimentation was intentionally withhold or delayed. However, the flexible, dedicated and patient-driven “food service” approach needs to be further evaluated regarding compliance rates and caloric intake.

However, retrospective data on nutritional intake are often uncertain and difficult to analyze. Therefore, since we know that compliance to nutritional intake is approximately 60%-70% in dedicated ERAS/ERP centers, it would have been a great help with a definition of the current ERP protocol. What was the standard re-alimentation protocol other than resumption of low-fiber or diabetic nutrition as mentioned in the article? Could the authors add the amount of calories, nutritiondrinks etc POD 0-3 (ie what patients should receive according to the protocol)?

We thank the reviewer for this crucial remark and gladly expand on the institutional realimentation policy. First of all, caloric counts were not systematically captured for all patients and therefore, this data was not available in the setting of this retrospective study. Patients benefit of a flexible approach (through dedicated food service) and get to eat a general, normal low-fiber diet (typically excluding fresh fruits and vegetables). In fact, patients may call and eat whatever they are hungry for, without caloric restrictions. The diet is complemented by nutritional supplements if needed.

Causality: We know that the surgeon decided in the end of operation who goes for ERP nutrition and who goes for individual nutrition (that is a strength with this study). However, individually treated patients had a way higher morbidity than the ERP patients and would probably have suffered from more complications regardless of nutritional protocol (confirmed by the multivariate analysis). No matter how you try in multivariate analysis, it is difficult to adjust for this, and find true independent predictors. Therefore, I think it is important to show exactly how the analysis was performed.

Which variables were used? The seven that are shown in Figure 2? Does probit regression mean that these variables were kept in the model all the time? That the current OR for each predictor holds for adjustments with the other 6 in the model?

First, we were interested in the question why surgeons would intentionally withhold a normal diet. Table 1 compared the two groups to look into this. Surgeons supposedly decided considering demographic and surgical risk factors and arguably, both groups were different. In a second step, we aimed to evaluate whether this “intuitive choice” was justified and tried to answer this question by including “ordering an ERP diet” in a model with POI as endpoint. All significant risk factors (seven indeed) upon univariate analysis for POI were included in the multivariable model. The probit regression model was used for the binary outcome POI (yes or no). The multinominal regression model adjusts for every other factor by providing odds for POI. We agree that several factors contributed to better outcomes in the ERP diet group. However, since ERP diet was independently highly protective, we believe that, similar to MIS, ERP diet should not be intentionally withhold based on surgeons’ choice and seeming demographic and surgical “risk factors” for not tolerating a normal diet

Minor concerns

Male gender seems to be an independent predictor for POI (at least when you look at the Figure) but 1.24 (0.89-1.32) is not significant as stated in abstract and text in the article.

We apologize for this typing error; the 95% CI was corrected in the abstract and manuscript.

I think the conclusion is too firm (considering the above mentioned in major concerns). Maybe the authors can make it softer??

We melted down the conclusion as suggested considering the above discussed limitations.